# COMSI^®^—A Form of Treatment That Offers an Opportunity to Play, Communicate and Become Socially Engaged through the Lens of Nature—A Single Case Study about an 8-Year-Old Boy with Autism and Intellectual Disability

**DOI:** 10.3390/ijerph192416399

**Published:** 2022-12-07

**Authors:** Kristina Byström, Björn Wrangsjö, Patrik Grahn

**Affiliations:** 1Department of People and Society, Swedish University of Agricultural Sciences, Alnarp, SE-230 44 Lomma, Sweden; 2Region Västra Götaland, Habilitation & Health, Children and Youth Habilitation, SE-541 50 Skövde, Sweden; 3Department of Children’s and Woman’s Health, Karolinska Institutet, SE-171 77 Stockholm, Sweden

**Keywords:** nature-and animal-based treatment, autism, children, joint attention, mentalization

## Abstract

This case study shows how an 8-year-old boy with autism and mild intellectual disability underwent positive psychological development in terms of play, social communication, and mentalization during a year and a half of group-based therapy using COMSI^®^-(COMmunication and Social Interaction). This eclectic treatment has a relational approach and is based on developmental psychology, knowledge of autism, and the impact of nature and animals on human health. The change in the child was been studied using both quantitative and qualitative methods. His general intellectual capacity was measured using the Wechler Preschool and Primary Scale of Intelligence, and his Mentalization Ability/Theory of Mind was assessed using three tests: Eva and Anna, Hiding the fruit and Kiki and the cat. Throughout the study period, change was documented with the help of the therapists’ process notes and the parents’ descriptions. The results show that support for the child comes from three different sources: nature, animals, and the therapists. Animals and nature form the basis for episodes of coordinated attention in conversation and play with therapists. The therapists’ approach used sensitivity and compliance with the child’s needs and focus of interest.

## 1. Introduction

Autism is characterized by core deficits in mutual social interaction and communication and limited variations in behavior, interests, and play [1]. Research has not been able to reach agreement on any specific theory that explains the underlying mechanisms of autism [2], but over the past 20–30 years, researchers have highlighted three primary areas of cognition that operate differently. The first is mentalization: the ability to attribute thoughts, emotions, desires, and intentions to oneself and others, and the understanding that these states affect human action [3]. The second is executive function (EF): the ability to plan and perform complex actions [4,5]. The third is central coherence (CK): the processing of information as a whole and based on context rather than details [6,7] Comorbid disorders are common, and 20–40% of all children and adolescents with autism also have an intellectual disability [8] Over 90% of children also show abnormal sensory reactions [9]. Problems with emotion regulation are common, as are high levels of anxiety compared to many other clinical groups [10,11].

Young children with autism have difficulties with the early milestones in the development of mentalization. These difficulties include limited eye contact, a generally lower degree of social interest, and a lower degree of visual attention to social stimuli, which are the basics of social communication [12,13]. Children with autism also have a limited ability for shared attention and mutual communication and pretend play [1,14], abilities that are also considered to be linked to the early development of mentalization in neurotypical children [13]. Today it is common in treatment to promote these abilities, seeking to create conditions for children with autism to develop their social abilities, language [15,16,17], cognition, and mentalizing abilities [18].

Interest in developing treatments for children with autism has increased in recent decades. Today a variety of models support the psychological development of young children with autism [15,19,20,21,22] with varying results [15,23]. Many treatment models provide support in a very structured way, but some have a more naturalistic approach and follow children’s own initiatives to a greater extent [15]. However, it is very difficult to estimate the success of treatment for children with autism, partly because the target group is very heterogeneous. There is probably no one approach that suits all types of treatment goals or situations [15,24]. Detailed case studies have been suggested as a way to present research with valuable knowledge about the best treatment approaches for individual children and can be used to supplement general evaluations [25].

This article focuses on a new form of psychological treatment in an early phase of development: COMSI^®^, which uses animals and nature as a platform that forms the basis for interaction [26,27]. The environment, together with therapists’ interventions, aims to create supportive conditions for children with autism—a kind of microenvironment [26,27,28,29,30] for psychological development through interaction. The original description of COMSI^®^ included an evaluation of its effects, which were found to be positive [26]. This article presents a detailed and in-depth study of one case; the research was conducted as a mixed-methods single-case study using both quantitative and qualitative methods [31]. Deficiencies in the individual methods used in the research are offset by using several types of research methods. If the results converge, this provides validation of the results. To the best of our knowledge, there are no studies where therapists use nature and animals to help provide support for the long-term psychological development of children with autism. This study, therefore, contributes to knowledge in the field.

### 1.1. Theoretical Background on How Nature and Animals Are Intended to Enhance the Treatment

In a recent article, Byström et al. [27] propose several mechanisms that may account for the positive effects of involving nature and animals in a developmentally supportive treatment for children with autism. They classify these into three key categories: (1) Reducing stress and providing peace and quiet, (2) Arousing curiosity and interest, and (3) Attracting the children’s spontaneous attention. The authors provide abundant examples of the above key categories. Daniel Stern’s theory on the subject of vitality [32] offers significant explanatory value for how the three key categories benefit children with autism and improve their cognitive functioning and ability to engage socially with therapists. Forms of vitality, according to Stern [32], are a type of movement perception interior to the person that accompanies all our experiences and gives them a dynamic form based on experiences of movement, force, time, space, and direction/intention. This is how vitality and the feeling of being alive emerges, Stern argues [32]. This perception of movement makes the content of thoughts and emotional experiences more fluent and gives rise to globalized experiences rather than just details. According to Stern [32], forms of vitality are also part of episodic memory, and even small movements can help people recall past experiences. The hypothesis we refer to here [27] is that in a therapeutic context such as COMSI^®^, nature and animals can stimulate particularly favorable forms of vitality for children with autism. Events that children perceive more clearly and with increased understanding and less stress and confusion may be particularly favorable for fostering positive moments of communication and social interaction between child and therapist. According to Stern [32], the specific qualities based on forms of vitality might form a kind of glue in the mind that binds thoughts, feelings, and sensations. Byström, Hägerhäll, and Grahn [27] suggest that this could have a positive impact on the weak central coherence that children with autism generally exhibit. Forms of vitality can provide peace and quiet and thus counteract stress, and help to increase the child’s interest and therefore facilitate interaction with the therapist. The sudden, unexpected events that are part of the dynamic experience of forms of vitality also increase arousal and hence alertness. In this way, forms of vitality may improve the conditions for more abstract thinking, such as mentalization, which benefits the ability to use language and engage socially. To accomplish this, children must engage in spontaneous initiatives and actions, and therapists must be sensitive and responsive to their needs and focus on their interests in the moment. It is also crucial for therapists to notice and take advantage of opportunities spontaneously occurring in nature and with animals present during the session in order to capitalize on them in the therapeutic intervention [26,27,32].

Forms of vitality are a new area to explore among children with autism, as a way to overcome their difficulties with being socially in tune and understanding social communication. However, some studies have shown that children with autism have difficulty noting forms of vitality in social communication with other people [32,33,34].

### 1.2. Nature- and Animal-Assisted Interventions

Therapeutic treatment and educational contexts increasingly incorporate nature and animals into their programs [35]. Natural environments have been shown to reduce stress [36,37], improve the ability to focus attention [38], increase curiosity, motivation, and commitment to learning [39], and offer opportunities for physical and emotional activity through play activities in nature [40,41,42]. Systematic reviews support the claim that introducing animals and nature into treatment and pedagogy has demonstrable positive effects [43,44,45,46]. Several studies have reported that therapy sessions for children with autism which incorporate dogs or guinea pigs have had the effect of increasing social initiatives, decreasing typical autistic behaviors, reducing stress, and lowering the children’s level of anxiety [47,48,49,50,51]. Treatment incorporating equine-assisted therapy was found to result in improvements in behavior, social interaction, and communication [52,53], with similar findings for therapy incorporating household pets [54]. However, we have limited evidence about how children with autism respond to being in contact with nature. In one recent study, Chinese parents reported that exposure to nature provided their children with motor-sensory, emotional, and social benefits [55]. They also identified several barriers, however, such as fear of their children misbehaving in public, safety concerns, and phobias. Another study presented guidelines for designing a sensory garden for children with autism [56].

### 1.3. The COMSI^®^ Model

COMSI^®^ is a one-and-a-half-year group therapeutic treatment encompassing nature- and animal-based interactions and communication aimed at children with disabilities, mainly autism, both with and without intellectual disabilities [26,27]. It is designed for children with a mental age of four to six years who have initiated speech development. The treatment environment is located at a small farm with animals and surrounding natural areas where children can participate in games and activities together with staff. Children attend once a week during school hours for sessions that last about two and a half hours, totaling 35 visits over the course of the program. The treatment primarily aims to increase children’s ability to engage in social communication and interaction and thus build up what Stern [57] describes as the important implicit knowledge of relationships: “how to be with others”. It is expected that mentalization, language, and cognitive development will be stimulated indirectly and directly as therapists help the children verbalize their experiences, among other aspects. Staff seek at all times to help the children understand and create meaning out of what is happening, and they follow the children’s initiatives to a great extent. The program team includes a psychologist who leads the work, and the whole team is under joint supervision by a psychotherapist on a regular basis. The COMSI model has been implemented previously, producing positive results in terms of play ability and interaction ability in the four children who participated [26].

COMSI^®^ [26,27] is an eclectic psychological treatment based on theories about the importance of nature and animals in psychotherapeutic and pedagogical contexts [35,42,51,58,59,60,61,62,63,64,65], as well as theories about development in neurotypical young children [66,67,68,69,70] and children with autism [3,4,6,7,18,71]. COMSI^®^ therapists work from a common point of view and approach, which includes being available, encouraging children’s own initiatives to engage in curious exploration, to play, and to be physically active [26,27]. In the therapy they share the children’s experiences, confirm their feelings and developmental stages, support them with emotion regulation, and assess the children’s direction in their development, so as to facilitate the next step [26,27]. Therapists often combine nonverbal and verbal means of communication in “joint attention” skills, including declarative pointing, glances, words, and phrases that call on and coordinate their attention and the child’s attention toward an object [26,27] in what is referred to as triadic communication [72].

In this study we primarily focused on increasing clinical knowledge about how children with autism spectrum disorders can benefit from the COMSI^®^ treatment concept. We sought to identify and isolate specific traits in children with autism that occur in different combinations and strengths, and measure how these traits were affected by the intervention. Using a case study as a method allows for a very detailed and accurate description of the child’s developmental stages during and after treatment [31,73,74]. The fact that the child’s development is expected to take place through a complex process of interacting factors in a natural context is another reason we chose the case study as our research method. Additional reasons include our desire to be able to contribute to the debate about the most suitable therapies for different subjects, an issue that has been particularly pressing in the field of treatment research [25,75].

### 1.4. Study Aim

The study aim was to describe the development of play, social communication, and mentalization in an eight-year-old boy with autism, mild intellectual disability, and major communicative delays during and after treatment with COMSI^®^. The data mainly consist of process notes, parents’ statements before and after treatment, and tests for false beliefs [76] before and after treatment. The process notes emerged from the therapists’ shared reflection on the child’s behaviors. These notes were written down in connection with each treatment session. We also report data on the child’s cognitive and language level before and after treatment as a way to better understand their development in the areas of interest.

The child selected for this case study has a particularly challenging autism profile due to his uneven cognitive profile with respect to verbal and nonverbal cognitive abilities, and he exhibited significantly better cognition capacity in nonverbal activities.

The study was approved by the ethics review board of the Sahlgrenska Academy at the University of Gothenburg, Sweden; Ad Ö 348–01; 23 October 2007. The child’s parents approved the publication of the research in the form of a case study. The child will be identified in this report using a pseudonym. Animal ethics has been guided by the commonly accepted ‘3 Rs’ [77].

All parents gave their informed consent for their child to be included in the study. The study was conducted in accordance with the Declaration of Helsinki [78].

## 2. Materials and Methods

### 2.1. Approach

The Mixed-Methods Single-Case Research (MMSCR) approach focuses on one individual subject. This approach may employ both quantitative and qualitative methods to assess the effect, feasibility, suitability, and meaningfulness of a treatment, program, or intervention for the individual subject [31]. The various methods used can then be triangulated to determine whether their results converge; agreement across the different study methods signals a high level of validity [79]. MMSCR is the preferred research approach for studying rare or unique conditions or when it is impossible to study a large, homogeneous sample of cases with similar conditions, as is often the case with children with autism. MMSCR is recommended, for example, for disability research on atypical disorders or for studies involving unusual comorbidities.

### 2.2. Participant

The group consisted of four children with autism and mild intellectual disabilities aged seven to eight years old, and one seven-year-old boy with an acquired brain injury and mild intellectual disability. All the children with autism had started to talk. From this group, “William” was selected as the study case based on the criteria of a diagnosis of autism, completing the treatment program, and having the lowest degree of verbal ability. During the COMSI treatment, William did not receive any other therapy from the rehabilitation center. The diagnosis of autism was confirmed one and a half years prior to the start of treatment, and the diagnosis of intellectual disability was given two years prior, when William was five years old. A psychologist and doctor specializing in autism conducted the autism assessment.

### 2.3. Course of Action

The participating children and their parents were prepared for the treatment in various ways: for example, the parents visited the farm, and the teachers informed the children and showed them photos of the environment and the staff. The parents were also given information about the purpose of the treatment and were encouraged to use photos from the treatment sessions that staff sent home regularly as a basis for conversations with their children. The letter with these photos was addressed to the children but also contained a letter addressed to the parents with a brief summary of what had happened during the session. These letters were intended to make it easier for parents to be good listeners and engage in conversation with their children. Before the summer break, parents and siblings were also invited to visit the farm along with the children receiving treatment.

Each child had their own therapist, but they were encouraged to seek out other adults and children in the group to interact with, something they did readily. After each treatment session, the therapists shared their reflections on the children’s behaviors and wrote down their process notes. The aim was to understand the children’s communication and behavior, as well as their progress in therapy. The psychologist, who was also the project leader, led the conversations and was also the one who compiled the process notes. This supervision and reflective work helped the staff to provide support to the children and to perform their professional roles. Once every four to six months, the project leader—sometimes together with each child’s therapist—met with the children’s parents to discuss the children’s experiences, development, and support. The aim was to create a consensus and optimize cooperation with respect to their children.

In between these meetings, the parents had the opportunity to contact the project leader if they had questions, and they were encouraged to do so and to make note of their children’s progress.

### 2.4. Assessment

Choosing a suitable assessment tool for children with disabilities is not easy. To get a nuanced picture of general intellectual capacity, the WPPSI-III, Wechsler Preschool and Primary Scale of Intelligence—Third Edition [80,81] was used, with all subtests being used if possible. This test was selected since it is one of the most widely used and tested scales of cognitive development for preschool children and has good evidence of validity and reliability [82,83,84]. One consequence of using a non-age-appropriate test is that the methodology for calculating the results changes. Instead of transferring the raw score to the scaled score, which is standard, a developmental age was used, with the help of a standard table for this purpose (A12). The table describes the norm group’s results based on the age when the group passed the test at the average level. The average developmental age was then calculated for the various sub-areas, such as verbal, performance, speed, and basic language. In general, the results reported are approximate and are used as a background variable. Nevertheless, we think they add an interesting aspect to understanding each child’s progress in therapy. The test of Reception of Grammar, (TROG 2) [85,86] was used to estimate grammatical understanding. This test is also widely used and has demonstrated good external validity and internal reliability [87]. Three tests were used to assess mentalizing or Theory of Mind: Eva and Anna, Hiding the Fruit, and Kiki and the Cat. All three estimate the so-called first degree of false belief, a mental developmental milestone that occurs at approximately four years of age and relates to understanding and interpreting the world based on concepts such as faith vs. knowledge and the difference between them [76]. The three tests have been used extensively for a long time, where not least the Sally and Anne test has been examined several times regarding internal consistency, interrater reliability and test-retest reliability, and has been shown to be reasonably reliable [88,89,90]. The selected tests are often used in research on children with autism. The reliability is good, especially if a composite is calculated using a number of tests [89,90]. However, the scores of children with autism on these tests do not necessarily mean that they can employ that same level of mentalization intuitively in real-world social situations. The tests differ in the degree of language comprehension required to perform them. One test (Hiding the Fruit) also includes an emotional component, where the child must predict whether the experimenter (Winnie) feels happy or sad, in addition to identifying a false belief [91]. Eva and Anna is a Swedish version of Sally and Anne—the test developed by Baron-Cohen, Lesli, and Frith [3]. The test is the least linguistically demanding of the three, while Kiki and the Cat is the most demanding, as it is administered in the form of a book [92] (pp. 464–466). The test of mentalization was administered by the psychologist/project leader. The WPPSI-III was administered by the other psychologist involved in the treatment program, while a speech therapist administered the TROG 2. The degree of autism was estimated after the first five sessions using the Childhood Autism Rating Scale, CARS [93] to get a clearer picture of each child’s difficulties but was not used at follow-up. The pre-treatment assessment was made approximately one month before the start of the program, and the follow-up was conducted approximately five months after the end of the program, for practical reasons (children’s summer holiday) and because we wanted the treatment to have as much impact as possible on the psychological development.

### 2.5. Process Notes

In this study, one of the authors participated in the treatment, evaluation, and reflection process via process notes. To validate the process notes, all therapists discussed their content until a consensus was reached. After each session, notes were made regarding the children’s developmental stages during treatment with respect to trust in the therapist, interest, and curiosity in exploration, emotion regulation, play, social communication and mentalization. Fonagy et al. [94] have developed a mentalization theory that describes different stages and dimensions of maturity (in particular we draw from the elaborations on this theory presented in Freeman [95] and Karterud and Bateman [96]. Since the psychological development of children with autism deviates from that of neurotypical children and they do not reach several of the milestones in the same predictable way, only certain parts of this theory was used. Proposals for and descriptions of the four different dimensions of mentalization are taken into account in the analysis of data [96]. These are (1) Automatic vs. controlled mentalization, (2) Internal vs. external mentalization, (3) Mentalization of the self and others and (4) Cognitive vs. affective mentalization.

### 2.6. Parents’ Description

The project leader conducted a semi-structured interview before and after treatment. The interview took place at the habilitation center with the participation of both parents, who were asked to describe their child’s personality and interests, ways of expressing basic emotions, ways of communicating and social interactions, reactions to sensory stimuli and possible fears. The follow-up interview started with questions about recent developmental progress and specific follow ups to the questions mentioned above. The interview explored both problems and progress in development. The interviews were tape-recorded and then transcribed.

### 2.7. Data Analysis

#### 2.7.1. Testing

Test results from the tests of general intellectual ability and linguistic ability are reported as a developmental age. The false belief tests are reported as pass or fail.

#### 2.7.2. Process Notes and Parents’ Description

After completing the treatment, the first author worked with the process notes and then focused on play and social communication. Trust in therapists, interest, and curiosity in exploration, as well as emotion regulation were the three aspects we chose to describe together in one theme, which we labeled basic preconditions for development. The reason for this is the assumption that the natural environment can have a positive impact on children’s ability to use more usual pathways for their social communication and learning that are more similar to the circumstances in which normal development occurs. For that reason, we use a subjective narrative form to describe the case child’s development, combined with the authors’ comments, to guide the reader in their understanding of what took place.

## 3. Results

### 3.1. Background and Early Development

William was eight years and nine months old when he began treatment. He lives with his mother, father, and two-and-a-half-year-old brother on the outskirts of a small Swedish town. Both his father and mother work outside the home. William started preschool when he was one year old, and when William was one-and-a-half years old, his mother noticed that she could not get him to make proper eye contact with her. He also did not develop speech as expected. When he could not make himself understood, he had violent outbursts.

William was very sensitive to changes in his routine at an early age. His parents eventually found a way to communicate with him using keywords. William had specific problems with getting haircuts and eating certain foods, and he disliked doing things with his hands. At preschool, William was described as seeking social contact with other children but quickly losing interest. To make use of his time at preschool, William was assigned his own pedagogical resource. William was diagnosed with an intellectual disability at age five, and at age seven he was diagnosed with autism and specific impairment of articulation. Attempts were made to assess his linguistic abilities in connection with the investigation of his autism, but the assessment could not be completed due to developmental age. The psychologist who conducted the autism study used the ADOS-2—Autism Diagnostic Observation Schedule [97] and noted that William spoke using two-word phrases and occasionally used three-word phrases. William was assigned a total of 19 points on the ADOS module 2 scale, on which the threshold value for an autism diagnosis is 12 points. William’s subscale distribution was 7 points for communication and 12 points for social.

### 3.2. Testing in Connection with the Start of Treatment

William was 8 years and 8 months old at the time of testing before the start of the COMSI treatment. His verbal intellectual level, as estimated using the WPPSI III, was calculated only on the subtest information, because this was the only one of the five verbal scale subtests that he passed. William was assigned a level of 2 years and 10 months on this subtest. His non-verbal abilities were estimated using the five performance subtests of the WPPSI III (block patterns, matrices, image categories, image completion, and figure composition), where he achieved an average developmental age of 5 years and 2 months. His developmental age on the test for grammatical language comprehension was calculated at 4 years and 6 months using the TROG 2. He did not pass any of the tests for mentalization.

### 3.3. Williams’ Abilities as Described by His Parents at the Start of Treatment

#### 3.3.1. Emotional Awareness and Emotional Stability

At the start of treatment, Williams’ parents described him as predominantly stable in mood and compliant at home. He clearly showed emotions such as joy, anger, and sadness by using body language. When he was happy, he acted somewhat playfully, when angry he could become physically active and when he was sad, he cried. He could also express his feelings, such as joy or sorrow, in short phrases or in single words, and sometimes even with some explanatory word. During a period when he was downhearted, he said “alone”, without giving any further explanation. His parents then thought he had a conflict with someone at school.

#### 3.3.2. Contact, Relationships, and the Ability to Share Attention

William wanted to be close to his parents and sit next to them but had a harder time dealing with the contact with his little brother. Misunderstandings easily arose, which also happened in relation to other children. If William’s little brother did not leave when he told him, he did not want to play with him and would throw things at him. His grandparents could sit and talk to him for short periods of time, but giving him a lot of attention was not effective. He was described as capable of sharing attention with his parents during activities such as watching TV or when they read him a fairy tale. He understood who won an automobile race and could state the name of the winner.

#### 3.3.3. Linguistic Communication

William’s parents had to use simpler language when talking to him, as he did not understand several sentences strung together in a conversation. Since his preschool years, his parents had heard him speak to himself occasionally. At home, he had learned to go into his room on these occasions. His parents thought he was talking to himself from experiencing stress earlier in the day.

#### 3.3.4. Interests

William was interested in computers and liked facts about things such as animals. The family’s cat and salamander interested him only a little.

#### 3.3.5. Routine-Dependent and Structuring Aids

William was only partially dependent on following routines at home at the start of treatment, and during treatment he had access to structural aids such as time aids, a yearbook, and pictures. William had just learned the names of the days of the week, which made it easier for his parents to prepare him.

#### 3.3.6. Concentration and Attention

William had difficulty in concentrating, which was noticeable when he had to learn things with the help of his directed attention capacity: for example at school or at home when at the computer. In such situations he was stressed by having another person next to him at the same time. He also did not like for other people to pay much attention to him. He did not want to participate in events such as Saint Lucia celebrations.

#### 3.3.7. School

When treatment started, William had just started second grade, but he had changed groups from a class of five students to a class with only three due to his need for more individualized teaching. The pedagogical approach was very structured—one thing at a time in a specific order, according to the schedule, with pictures as support.

#### 3.3.8. Sensory Problems

William was very sensitive to sound and light and did not like getting dirt on his hands; when he was outdoors he wore gloves. He had a hard time getting a haircut.

### 3.4. Progress during Treatment

#### 3.4.1. Basic Prerequisites for Psychological Development: Trust, Curiosity, and Interest in Exploring the Environment, and Regulation of Emotional States

During the initial treatment sessions, when William showed signs of insecurity, he said he wanted to go home or asked for his father to hold him. This meant that he had access to internal representations of security in the form of his parents, and these seemed to guide him as he carefully took the therapist’s hand. However, this contact was short-lived. As the first weeks went on, he sought out his therapist for protection and care. In the beginning he also had a hard time balancing his energy. His mood changed between happy, alert, and curious to introverted, low-energy, and uninterested. The way he dealt with fatigue was similar to that of an infant: he would turn away for a moment and then resume dialogue with his mother with renewed energy. William also turned inwards but re-engaged after a while with renewed strength and was then more available for contact again. William was positive about the treatment environment from the beginning and focused his attention on the animals he saw. He did not want to get dirty and always wore gloves. He usually enjoyed riding the horse and also liked to play with the dog; he also noticed birds flying past. He gathered eggs from the hens together with staff and observed the roosters roaming freely in the yard. He showed particular interest in the rooster named “Crossbill”. He was fascinated by the rooster’s disfigured beak and expressed joy and wonder at it, which he shared with his therapist, even making eye contact. At first William was not very interested in contact with the other children; he did play alongside one child but not with him. When he sat by the fire, however, he was more engaged, and his ability to switch from one activity to another also increased. He could then switch between being part of an activity with his therapist, such as “blow on a glowing stick taken from the fire”, digging all by himself with a stick in the ground, or just resting. He seemed to enjoy the time and to be feeling well. The location around the bonfire with the therapist sitting next to him seemed to add to William’s mental energy and increased his motivation to do things. He got ideas about things to do. His ability to shift and coordinate his attention with the therapists also seemed to be positively affected.

Through the support from the environment and his therapist, William took further steps in his development after he had been in the treatment program for a couple of months. He started looking for new things that aroused both curiosity and a touch of fear, which gave him the opportunity to practice regulating his emotional reactions. For example, he showed a fascination with spiders and ants that he had looked for, but he was also a little scared of and angry at them. At first he wondered whether the ants were “looking at him”. This is interesting, as it shows that William actually thought about an insect’s behavior and possible intention towards him. William seems to be investigating what happens when animals and humans meet; he does not know the answer, so he asks the therapist, whom he thinks has an answer. He certainly knows that eyes can be used to look at others, and perhaps he also knows that eye contact can precede actions of various kinds and that they signal intentions. The question is whether he is also thinking about what the ant intends to do next? Maybe it’s going to bite him? By having the therapist near and listening in, William is led to mentalize verbally, and it probably helps him to regulate the emotions that may have been evoked.

As time went on during the first treatment period, William became more and more alert and happier, and he increased his ability to signal his needs and feelings verbally. For example, he could say how he wanted others to behave towards him. On one occasion, he told a child that he was not allowed to hug him, and on another occasion when the other children wanted to climb a small hill, he said no but still followed along the walk. He also became more tolerant of reprimands and boundary setting, and not only did he express his feelings more clearly but he could also deal with adversity and became less impulsive. It seems that William now gained in self-assurance and self-understanding and achieved more vitality. The ability to regulate closeness and distance to the other children in the group was also improved. Perhaps a certain increase in cognitive capacity could be one reason for this. Once, in the middle of treatment, as he walked past the anthill, he said he wanted to destroy it, but suddenly he changed his mind because he “felt sorry for the ants”. This compassion was evoked spontaneously and led to emotional self-regulation. This mentalization was verbalized explicitly and contained emotional content. He could contain his emotions and thus inhibit an impulse to destroy. He also had a purpose in his communication with others, which was to make the social contact function better. This indicated an increased interest in social contact, and he might have been developing a need to belong to the group, albeit on his own terms.

Williams’ increased ability to express himself verbally and to regulate his contact with others in the group went hand in hand with him becoming more present and no longer so guarded. Sometimes, however, he could be a little impatient and frustrated, a tendency that reached its climax shortly after the start of the final semester. It seemed as if he needed some kind of new stimulus. To meet Williams’ needs, after a conversation about how he felt, he was offered his own small bonfire to take care of—a suggestion he immediately accepted. After that, he became more satisfied, harmonious, and compliant in his contacts. For example, he could now maintain a level of enjoyment in games that were a bit more challenging, such as play fighting or snowball fights, he could join in games that were already underway, and he showed interest in mischievous games that he had previously avoided (for more examples, see the theme of play). Once again, fire becomes central to Williams’ self-development. To be able to take responsibility for his fire, put firewood in it, and keep it under control, after gaining an increased understanding of his own emotional states via the therapist’s affirmative words, seemed to be especially satisfactory. Now followed new developmental steps regarding self-regulation during social interaction with the other children and the therapists. He grew in competence and regained additional mental energy. The timing of the task seemed to be optimal; William was ready to take on more responsibility—both concerning the regulation of his own inner states and behaviors and the regulation of the external source of warmth, his own bonfire. This growth process went on during the last weeks of treatment, during which further developmental steps were taken in various areas, especially with regard to verbal communication (for more on this topic, see the social communication theme). The length of his contact time with therapists also became longer, and he could recount events that had happened previously, talk about things he was afraid of, and ask for things he wanted. He seemed more certain when he expressed his opinions. All of this progress further improved the level of contact staff could have with him. He was no longer guarded.

Towards the end of the treatment, William sought out one of the other boys, who experienced mood swings and whom William had previously avoided. Now William imitated him, teased him on a few occasions, and withstood a surprising amount of teasing in return. Teasing is an example of a higher level of mentalization, which requires good knowledge of another person’s emotional reactions and what causes them. Teasing is also a way to train a higher level of mentalization, by reading the other person’s inner states and at the same time expressing oneself verbally in a way that should have a certain clear consequence, which you then must be able to deal with yourself. This indicates that William now trusts himself to handle more aggressive impulses without the situation getting out of hand. He is not afraid to use aggression within the framework of teasing verbally, in a way that is playful and not serious. William used to be afraid of ants, but by the end of treatment he showed an anthill to one of the other children. He held out a stick with ants crawling on it and asked, “Are you afraid of ants?” He himself was calm and not afraid. One may wonder if this act is about William wanting to show the other child his new skill of bravery. In that case, it could be a sign of an underlying increased social need. It could also be that he now is curious about how the other boy will react, what he will say, feel, and think. In this case, he would be showing mentalization about another person’s inner states, using both a gesture (reaching out and showing) and verbal expressions.

#### 3.4.2. Ability to Play

At the beginning of treatment, William played alongside other children, but after a couple of weeks he started playing a bit with other children in simple games such as chasing or shooting each other. He also played simple fantasy games on his own and let the therapist listen. For example, one time he sat digging in the ground with a stick and said that he was building a house for a mouse and that everyone must be quiet so the mouse could sleep! This can be an example of what Fonagy and his colleagues [94] describe as a pretend mode in the child’s mental development at the age of three. Now there is a boundary between inner experiences and outer reality, which is not the case in the previous level, called equivalence. The child can now take refuge in pretend play but may at first have difficulty moving between the levels of pretend and reality. The game can easily be destroyed if it is commented on incorrectly. Maybe that is why William is so careful to give us an order to be quiet and not disturb. Pretend play is an act of training to imagine things that are something other than they really are. To do something in the mind is a form of mentalization. William managed to play hide and seek right from the start; he told his therapist, “You count for playing hide and seek” and hid really well so as not to be seen. It shows he is certain about the idea of the game, of the importance of others’ visual perspective of him. One aspect of the game that seemed especially amusing to William was to run away and hide—i.e., a form of ingenuity towards the adult—and he liked the moment when he was found. In early childhood, being found is a moment of emotional reassurance and helps to build trust between the child and their caregivers. In this context, William expressed a lot of positive affect in his communication at that moment, both verbal and nonverbal. William also enjoyed more physical play, such as when his therapist spun him around. He also shared joy with his therapist in more developmentally basic forms of play, such as peekaboo. When the hat went down, the therapist said; “Now it has gone down”, at which point William laughed and turned his face away, then faced forward again and showed that he wanted to continue playing the game. This game gave William the opportunity to practice shared attention, taking turns, and coordination in dialogue, such as listening to someone else’s signals and being listened to, and finding the rhythm in the turns. Peekaboo is a very common game in the dyadic interaction between young children and their parents that provides both affirmative emotional support and valuable training in social communication.

Most of William’s games at the beginning of treatment concerned regulating distances to other children, adults, and animals (especially the dog), as well as creating his own ideas, which he puts into action. William liked games that involved fast movement, where the excitement builds up and wears off. In this kind of game, he gets a lot of experience in learning how to regulate his own feelings and behaviors, coordinating his attention with others, and how to be part of a play context with others. If there was too much tension, he ran away, but if he wanted more intense contact, he came close again. The games involved large movements, and there were no clear rules for how to behave or what to do—only approximate expectations of what would come next.

After a few occasions, William became interested in participating in roleplaying “cops and robbers”. He did not understand the basic structure of the game at first, but was interested and got engaged. When one of the therapists asked William to unlock the door to the “jail” where she was being held by another child, he did as she told him to and imitated the other child’s hand gesture. Both boys then locked her in and unlocked her, and after a while they helped one another in a mutual act to unlock the jail cell and set her free. In the following sessions, William increased his engagement in social interaction with the therapist; his face shone with joy, and sometimes he took part in various play constellations with other children, including one quite lengthy sequence with one of them. In the thrills of the “cops and robbers” roleplay, William used new elements of pretend. He soon understood the simple structure of the game (tracking, catching, imprisoning, unlocking, and sneaking), and he watched and imitated another child’s hand gesture of how to use a pretended key and was able to use the gesture himself. As he became more confident in the roleplay, he became more enthusiastic and emotionally engaged, and he got closer and closer to the other boy. At times they worked as a team that collaborated against the therapist. Eventually, William developed his play capacity in terms of variations, duration, reciprocity, independence from therapists, and taking part in group play. When playing cops and robbers, he participated in several new elements of the game, such as sneaking away together with the other boy after William unlocked and freed the therapist. On one occasion, he joined the play of others. To be able to join an ongoing game requires more complex mentalizing capacity, coupled with planning (executive function) and the ability to act (knowledge of the rules of the game). In this example, William’s mentalization required interpreting one’s own wishes and those of others, thinking about what happened in reality in the play scenario, and when to jump in. After a couple of months, he and another child went on their own out to the fields nearby, where they chased one another in and out of puddles of water, in a game that lasted longer than William’s previous play interactions. This game of chase, which took place without the support of therapists, involved social contact and a buildup of tension that was colored by sensory input and the dynamic elements of running in the water.

From that point on, William’s game participation became more filled with meaning, and new games emerged, such as throwing a ball to the dog, playing catch with another boy without the help of therapists, and playing hide and seek with another child. William also starts telling staff about his plans. At one point, a month later, he loudly declared what he was going to do: “I will shoot the bull moose today”. At the same session, William played with three other children in a game that involved hitting a rock with a stick. The game lasted for ten minutes, which was a long play scenario without adult support for this group of boys. They said they “shot a bear”. At the next session William participated in a snowball fight with the group of other boys and did well at it. Now we see that William has developed a new flexibility and ability to vary his behavior in games with other children. He enters a fantasy of being a hunter and inserts new play scenarios into that fantasy. Williams’ motivation for and ability to tell the rest of us what he wants to do has also increased.

Another new game one of the therapists introduced involved magic and turning others into wild animals. The structure of the game required participants to be aware of different roles: for example, being the one in control or the one who had to follow someone else’s initiative. William played that game happily with other children, both with and without the support of his therapist, and he managed to both “transform” other children and therapists, petrify some of them, and turn them back into people or to the chosen animal. This game also requires a more complex mentalizing ability and planning ability. It is important to be able to interpret the magician’s verbal and nonverbal expressions, carry out his commands, and also be able to come up with animal names and choose who in the group is to be the animal in question. At one therapy session William was especially full of vitality, and during the ride up to the play area in the woods he exclaimed with great enthusiasm, “We are going to play ships and whales!” Upon arriving, a period of fantasy play started at a spot with a big rock, which was largely led by William. The game involved “catching sharks and whales”, and William enjoyed sitting on the rock helping the therapist “save” the others by pulling them up on the rock. At this point William had taken on a new leadership role in the games, and he both invented a fantasy that works well with the group and a role for himself that he was comfortable with and could handle. During the last treatment period, William further increased his ability to reciprocate in communication with others during games, and he also became a little better at initiating games. In snowball fights, he initiated with the help of his gaze, which signaled the right kind of expectation and joyfulness. His ability to keep games going had also increased. Now William shows clear evidence that he can use implicit forms of mentalization, i.e., to unconsciously and automatically use his gaze to signal anticipation or to color messages with emotional value. It also shows an increased awareness of the importance of social signals as a tool in interaction with other people.

#### 3.4.3. Social Communication Skills

At the beginning of therapy, William could only listen very briefly. He used single words or short phrases and only managed one or two turns in dialog. Eye contact was also short-lived. During reflection work, the therapists compared Williams’ way of relating to a butterfly’s, light, fleeting contact. On the other hand, he commented on some of what he saw and experienced and could put feelings into words, such as feeling cold and longing for his father when he was a little sick. The level of speech agreed well with the parents’ description and results from the tests.

When William rode the horse, his speech became more fluent. He repeated the phrases “I one” or “I two” and associated the experience with the car races he watched on TV with his father. However, he did not seem to care whether he rode first or arrived at the clearing in the woods first, but he looked proud and happy and made a victory gesture with his arms lifted after dismounting the horse, as if he were standing on the winner’s podium. The rhythmic stimulus of riding seems to bring up an earlier similar experience when he was together with his father, and he is guided by the internal working model when he expresses a new gestalt of how to be a winner. Such generalizations are quite unusual for children with autism. The implicit relational experience of being with his father sharing much joy in their mutual intertest was probably encoded in the episodic memory with the help of what Stern (2010) calls dynamic forms of vitality, and that can, at least hypothetically, explain why he could retrieve it quite naturally in a similar situation, being on a horse in treatment.

William told us when he wanted to be by himself and talked about what he did not want to do. At one point he said to a boy, “I do not want to ride, do not want to be with the others…want to be by myself”. On another occasion, he told one of the therapists to “go away” as she approached him, but then he quickly changed his attitude when he saw a spider, commenting, “Look, here comes a spider”, his initial impulse to ward off any attempts from the therapist to make contact replaced by curiosity.

We can see that William uses language to reject other people’s attempts at contact. He is not comfortable with having others too close to him, and he does not want to be part of the group activity of riding together, but when the spider appears and William becomes fascinated, there is a turning point from strong rejection to a positive approach when he gets an impulse to want to show the therapist the spider.

When William sat by the campfire, he used certain words and phrases drawn from fairy tales, but he could not enter into conversations other than to give brief answers to the therapist’s follow-up questions. For example, he might suddenly blurt out, “the hundred-acre wood”. He also narrated on the theme of death: for example, “The bear is dead and mother…lion…but the lion cubs are alive”. Over the course of the treatment William returned to this theme in various contexts. In connection with the death of a horse and finding skeletal parts of a moose, William talked to the therapist about what had happened: “It was a skeleton…. The moose dead…. It was a magic forest”. From then on, he called himself a “forester”, and his interest in finding animals and going hunting increased. Sitting still by the campfire together with his therapist makes it easier for William to get in touch with his inner self in a deeper way, and he then begins to narrate in a simple way based on thoughts and emotional impulses that had come to him.

Eventually William started to ask questions, and he also became more and more communicative. In the middle of the first treatment period, he went to the therapist and expressed frustration over another child’s behavior: “He destroyed my hut”. William’s communication continued to develop further, with turn-taking becoming more frequent and the content of his speech more coherent. He could also more easily switch between fantasy and reality. When one child asked who made a hole in the anthill, William first answered “the lynx mother”, but when the therapist said no, it could not have been that, he changed his mind and gave a more realistic answer: “the European woodpecker and the woodpecker”. One example of a more extended dialog was when he went for a walk in the woods with his therapist and looked for animals. William said, “Go to the forest, will look for animals”, at which point the therapist asked, “Is there anything more in the forest?” William replied, “It’s a monster troll”, and the therapist asked, “Are they dangerous?” William replied with a laugh, “No, they are kind!” All of a sudden, he was taking six or seven consecutive turns in the conversation. When he stood by the moose jaws, he commented, “The moose is dead; poor moose, did the farmer shoot him?”

William’s verbal communication develops continuously. At first, he often used communication to signal that he did not want much contact, but after a while the content became colored by his experiences from being in the natural environment. He also begins to show joy based on the therapist’s comments to him. He uses his mentalizing ability when he consciously switches tracks from fantasy to reality when talking and when he thinks about and feels compassion for the moose’s fate, something he also expresses with words.

After a couple of months, William began to think about himself. At one point, he smiled and held up a piece of ice in front of his face while asking the therapist to take a photo of him. His parents got a copy of the photo and put it in a frame, which William really enjoyed looking at. In conversation with William at that point, we began to experience a deeper sense of contact and a better mutual understanding with him. It primarily shows that William now gets in touch with an inner need to portray himself, and he gets the idea from experiencing ice. It is unclear whether the wish is the result of conscious mentalization or not. The action, though, leads to a creative act of self-confirmation that gives satisfaction both to him and also to the therapists and his parents, who feels that William conveyed to them an important feeling about himself. Perhaps it was about how to be William with autism. At home, William was now starting to talk about himself as being different. In therapy he also began to ask for things he needed, such as water when he was thirsty. He also put new feelings into words, such as being tired, and he told us about things that had happened at home. Towards the end of the therapy, he asked more fact-oriented questions: “Does it hurt the cow when the calf suckles?” He also commented on the animals based on their family roles, saying, for example, “mother pig”, “they have teats”, and “father and children”. The form of mentalization William uses is about the animals’ feelings, and he knows that the therapist is a person who can give him answers to his questions.

In the final part of the treatment, William began to tell staff what he planned to do. For example, one day he said, “Hello! Today I’m going to look for animal skeletons!” He began commenting on things that had just happened: for example, when he told another child, “The dog licked my nose”. In general, he seemed more relaxed in communication. He had, for example, a playful turn-taking in dialog with the therapists about words and nonsense words. He also expressed his thoughts about what the mother cow and her calf were thinking. He asked the therapist what the mother cow thought of her calf, and said, “The calf dreams that he is flying”. At the end of treatment, his conversational content also became closer to reality, and he exhibited a greater degree of understanding than before. William’s increased communicative and social ability at the end of the treatment could be that he now better understands the important role of language in establishing and maintaining social interaction with other people. It could also mean that he now appreciates the type of in-depth contact that can follow from that. The role of language as a means to share thoughts and feelings and to stay in contact socially with other people is very difficult in autism [98]. This group of children usually uses language as a way to ask for things they want and not primarily to be in a relationship sharing thoughts and feelings with someone else. William also shows evidence of thinking about someone else’s thoughts, but in the animal world. This is a higher level of mentalization that he has not used before.

### 3.5. Parents’ Post-Treatment Description of William’s Progress and Development

#### 3.5.1. Emotional Awareness and Emotional Stability

William’s mother stated that she thought William had developed his personality a great deal. She said that William had removed stones from the wall that surrounded him in relation to other people. He felt more present to her and told his parents what he wanted to do more than he had previously. He had also begun to choose new activities.

#### 3.5.2. Contact, Relationships, and Ability to Share Attention

William achieved a closer relationship with his grandmother and grandfather, and he liked to spend time alone with them, something that was new and important to him. He was happy to share experiences with the staff. His mother explained that if he saw something interesting on TV, he would go and fetch his parents and wanted them to sit and watch with him. He had not shown interest in doing so previously.

William had also further developed in his role as a big brother. His understanding of what his younger brother could do had increased, and he could now wait for him more patiently. They both play a lot together: for example, kicking a ball, bicycling, bouncing a basketball on the garage floor, and doing things on the computer, and their parents could hear the brothers chatting about movies they have watched on TV or DVD. Occasionally they would get into quarrels with one another, usually about the younger brother wanting to start certain games that William did not want to play. He would then politely say “no” several times, but if that did not work he would get angry. When William thought something was funny, he would seek eye contact with his brother, and if he does not understand the plot of a film, he would ask his brother what had happened so he could understand it. William was at this point also thinking a lot about one of the teachers at his school who was on holiday on the other side of the globe. He said he missed her and wondered what she was doing; he wanted to talk about his thoughts and feelings with his parents. He also said that he was afraid that something would happen to her if she went diving among sharks.

William’s parents felt that the whole family was important to William: mother, father, younger brother, grandmother, and grandfather. When his father’s sister was in the hospital, he was worried, and his parents had to explain what was happening and that she would be fine.

#### 3.5.3. Linguistic Communication

William’s mother reported that William’s verbal ability and language comprehension had developed and that it was possible to get something out of having a dialog with him after the therapy. She further described that it was now possible to get through to William more directly: for example, if she told him that they were going to the bowling center, he immediately would ask in response whether they would also have coffee there. He had also begun to reflect on different topics. When he and his parents walked past a hotel called the Carpenter Hotel, he read the word and commented that carpenters stay there, showing that he recognized the word *carpenter* and drew a conclusion from it. He could also narrate and recount events that he thought were frightening. When he did not dare to go to the cinema with his school because of his sensitivity to sound, he and his teacher were able to talk about the whole event and together figure out exactly how to overcome all the obstacles one by one; after that, he himself began asking his parents for earplugs. Another situation that his parents reported as being resolved with the help of language was his issue with haircuts. William has always found them difficult, and there had been conflicts and quarrels before on the topic. Now William could control the pace that his mother cut with help from his younger brother, who told him a fantasy in which the younger brother suggested to William that the hair tufts were ghosts that he sprayed away. In this way, William was guided through the haircut until it was finished. His mother also recounted an important new event involving William that happened in direct connection with this successful haircut. Afterward the family went out for pizza, and suddenly William said that the pizza was especially nice and that he had longed for just that taste. He told his parents that he recognized the taste of a pizza he had eaten before and commented on it. His parents said he had never made that kind of comment before.

#### 3.5.4. Interests

William’s parents reported that he had started to play ball games often, both basketball and handball. He had become interested in newspapers, but probably reads the TV schedules the most, his father noted. Something new was that both children got their own kitten. William was very afraid that the kittens would disappear, and he wondered whether a fox could grab them. His parents reported that he picks up the cats in his arms and cuddles and talks with them and that he feels safe having them in bed for a while when he goes to sleep. They said that he had not taken on care responsibilities, but he did ask his parents if they had made sure to feed the cats. Sometimes he would tell his parents that they were not allowed to be the ones to give the cats their food, because he felt that his cat would then leave him, and he did not want that to happen.

#### 3.5.5. School

At the post-treatment interview, William’s parents reported that school was going well and that William had learned to read, could read a digital clock, and had started to learn English.

#### 3.5.6. Continuing Difficulties and Problems

William’s parents reported that he was still not interested in playing with his classmates after school unless his little brother was included. He still did not like to attract other people’s attention. At one family party, he sent out his brother to introduce himself instead.

### 3.6. Results of Post-Treatment Testing

#### 3.6.1. False Beliefs

William was 10 years and 8 months old at the time of the post-treatment assessment. In this assessment he passed two of the three false-belief tests. The test he failed is the most linguistically challenging: Kiki and the Cat. He also failed to attribute feeling sad or happy. In total, he scored 4 out of 6 points, compared to his score of 0 out of 6 prior to treatment.

#### 3.6.2. Cognition

William’s verbal intellectual level, estimated using the WPPSI III, was calculated using two subtests—information and similarities—and in the post-treatment testing he achieved a total developmental age of 4 years and 8 months. His nonverbal ability was estimated at an average developmental age of 6 years and 8 months, but he had reached the ceiling for the subtest matrices and image completion.

Comment: regarding the general intellectual level estimated using the WPPSI III verbal subtests, William advanced almost two additional developmental years compared to his scores prior to treatment, on the nonverbal subtests he had progressed by approximately one and a half years. This is a good achievement, given the fact that William also has an intellectual disability in addition to autism. However, it must be borne in mind that the test results from a few subtests in the verbal area are difficult to interpret as a measure of global verbal intelligence

#### 3.6.3. Language

His grammatical language comprehension reflected a developmental age of 5 years and 6 months, estimated using the TROG 2, which means a progression with one developmental year. Even though this also is a good achievement, well in line with his results on the verbal subtests of the WPPSI III in the post-treatment testing, he has still major language deficits that prevent him from expressing himself.

## 4. Discussion

Using an MMSCR approach, we have studied the development of play, social communication, and mentalization in William, an eight-year-old boy with autism and mild intellectual disability during and after treatment with COMSI^®^. The areas of development that showed the most progress are trust in others and in himself, emotion regulation, interest and curiosity in exploration, play, and social communication and social interaction. In addition, we placed special focus on tracing his developmental steps in the direction of increased mentalization ability, both nonverbal and verbal. Play and social communication are skills that are recommended for inclusion in treatment for children with autism because they are believed to create the conditions for the development of language, social skills, cognition, and mentalization [12,13,15,16,17,18].

The results of treatment show that William developed his ability to play, communicate, and interact socially over the course of the treatment when compared to his previous abilities, and his parents’ descriptions agree with this conclusion. The results of the tests also support the conclusion of positive development. His progress involved basic nonverbal skills such as using eye contact, gestures, and body language aimed at social communication, as well as verbal communication skills. After the treatment he narrated more coherently, offered comments, and showed the staff more things that he wanted us to see, and he started to use concepts about mental states that point to an increased ability for mentalization. For example, on one occasion he thought about what one of his teachers at school might be doing on her holiday, and on another occasion he and his therapist elaborated a story about a calf dreaming that it was flying. His executive function also seemed to have improved quite a lot, as we saw, for example, when William started to speak about his plans for what he wanted to play or when he wove in new play scenarios while play was in progress. His ability to regulate his emotions had also developed, and William was, at the end of the treatment, able to play more challenging games without the help of his therapists and used his gaze more often to signal his expectations and intentions in games. In other words, he had generally become more confident in social interaction, something that seems to be well supported by his improved understanding of context (central coherence), emotion regulation, and cognitive ability.

William also evolved better empathic skills, caring for other people and animals. At the beginning of the treatment, he did not show care toward the family pets, but afterward he showed a lot of care and great interest toward the new kittens. He had also matured emotionally: for example, in his relationship with his younger brother, with whom he now had more mutual contact in games and conversations. He has more realistic expectations about what his younger brother could or could not do, and he adapted better to his abilities. Overall, William seemed to understand more about what is happening around him after treatment. His mother said that it was possible to have more direct and mutual communication with him after the treatment finished. He could also more easily choose what he wanted to do and had started playing basketball and handball in his spare time. Overall, the result shows positive developments in his personality and improvement in specific areas of difficulties for children with autism.

The results of this study indicate that COMSI^®^ worked for William and that the treatment itself contains some specific features that we suggest are important in the treatment of children and adolescents with ASD.

Animals and nature form the platform and a microenvironment in which the treatment is based. Research on the importance of nature and animals for children’s development has found that various natural elements and processes stimulate children’s interest, imagination, and emotions, and that this supports associations, thinking, language, and the ability to reflect [42,99,100]. At present, there are no studies examining whether this also applies to children with autism, especially in terms of stimulating their capacities to play, communicate, and interact with other people with more engagement. Research shows that children with autism have difficulty with social contact, which also applies to people who suffer from severe life crises or severe stress. According to Supportive Environment Theory (SET), people under stress may find certain natural environments easier to relate to and manage [63,101], This applies to natural environments that are not too rich in impressions, and where a person can find ways to retreat and be by themselves when needed. According to SET, nature contains qualities that involve a gradient of challenges. COMSI^®^ is a method with a fixed structure. There are safe routines, starting with arrival at the farm for further transport to the campsite in nature. This natural site offers both a feeling of safety and a gradient of challenges, which makes it suitable for children with different needs. Here, however, it is important to point out that the children in the therapy group were at approximately the same level of cognitive and linguistic development and had approximately the same severity of autism (approximately moderate level on the Childhood Autism Rating Scale). Thus, it was easier for the children to interact more with each other over time and not become too hesitant or afraid due to a lack of understanding of what was happening in the interaction or because the environment and activities became too demanding.The therapists work to be responsive and provide gentle support. Staff see children with autism as “islands on the nature platform”. These “islands” are independent individuals who gradually receive support in building networks of interactions with the environment, where nature is the primary component. The children are individuals with strong integrity. They should be allowed to be at peace if they so wish and need. Their confidence in themselves and their surroundings needs to be built up, step by step, and this requires that they be allowed to explore the surroundings on their own terms to the extent possible. The therapists are trained to develop an intuitive sense of each child’s needs. With the help of gentle, flexible support from the therapists, children’s interactions with the environment can be developed. When the therapists discover that a child is taking positive steps forward in their development—for example by showing interest in something that is happening in the environment—they step in to help the child experience shared joy, coordinated attention, express their thoughts and feelings about what is happening. When they discover that a child cannot understand and interpret something they have seen or experienced, the therapists also explain and provide support. However, they step back to let the child take over again when they seem to be able to cope with the situation. We suggest that this nuanced and more well-toned social experience from being with therapist in nature, could have an effect on inner representations to emerge more easily, with Stern’s terminology so called RIGS (Representation of an Interaction that has become Generalized) [70]. Stern’s concept of RIGS refers to the way in which young children’s real experiences of interaction are organized in their psyche. Stern connects the emergence of RIGS to memory structures that represent such experiences. Children can begin to recall these models of “being with others” early on in the form of structures similar to stories. According to Stern, these stories contain different components, such as sensory impressions, actions, affects, and goals, and they form a temporally and thematically coherent unit. These experiences of interaction can then be generalized and form RIGS. One example of such a RIG could be “walking in the woods” under safe conditions with therapists. Models of being with others can thus start with being with nature.Children’s self-development follows two main trajectories [102]: The pursuit of autonomy—being able to understand and master the world around one to avoid falling victim to unpredictable or unknown forces—includes the pursuit of knowledge, the desire to create, and mastery of everyday situations. The pursuit of homonymy means belonging to a group, family, work community, sports association, or any other form of social community and can also include a type of belonging to a religious group or nature.Nature, coupled with a therapist who serves as a safe haven, can thus be a place where children can exercise autonomy but are also able to develop a type of necessary security and belonging. This is called place attachment and can develop when the conditions of person, place, and process are right [103]. Being able to develop a sense of togetherness and security in certain places is also considered a part of children’s natural development [104].Collaboration with parents both before COMSI^®^ treatment and during and afterward builds a bridge between therapist and parents. Children can sense this bridge of hope and trust, and it can also help the method to work. It is also important to assemble the group together correctly. Our experience here is that group members need to be at approximately the same developmental level for interaction between them to optimally stimulate their functions and for development to progress satisfactorily.

Previous studies have reported interesting conclusions [26,27] regarding what nature and animals may add to the treatment program, making it a good microenvironment for contact and social interaction, a short summary of which is included in the introduction here. These previous publications highlight forms of vitality [32] as particularly important in explaining the participating children’s increased ability to absorb and understand the treatment environment. The forms of vitality in the children’s experiences may have been more efficiently recorded in memory, leading to increased vitality and mental energy and improving their cognition and affective functions; for example, they might have demonstrated a normal tendency to integrate separate elements into a coherent whole (central coherence), along with executive function and mentalization. With such improved encoding of representations of vitality in their memories, children can more easily recall these memories and talk about them. Today we have several theories from environmental psychology [58,59,105,106,107] that highlight and show the different ways that people, in their long coexistence with nature and animals, have found and adapted ways to effectively interpret information regarding events in natural contexts. This may also apply to children with autism and help them to feel less stressed and calmer, and to show interest in and be able to focus on what is happening in the environment, spontaneously and curiously. If this is true, it may explain why the children in this treatment program were able to build relationship-based knowledge via socially engaged interactions with therapists regarding what was happening in the environment. Some examples of particularly positive episodes witnessed during treatment sessions include times when animals acted as bridge-builders between the children and their caregivers, which made it easier to establish and maintain contact and trust between them. Likewise, a wealth of episodes in the natural environment stimulated the children to create meaningful contexts and situations that they wanted to show and tell about. These include moments by the campfire where they were able to rest and relax, time in the forest where they were able to explore and experience events that aroused their imagination, and where they were able to listen to natural sounds such as birdsong and rustling wind.

We can summarize our interpretation of the process as follows: the environment challenges children at the right level—for example, experiencing bad weather, occasionally finding wild animal tracks, or encountering ants and spiders. On such occasions, the children turned to their therapists for help and support, which offered good practice in learning to regulate their emotions and behaviors more independently over time. All the physical movement that the treatment entailed—everything from riding to running and playing in nature—was also important as an element that promoted thinking and interaction. Through the specific qualities that nature and animals added, the children were able to absorb a breadth and wealth of positive interactions, which we believe increased the preconditions for optimal so-called experience-based psychological development. Through RIGS for “how to be in nature”, they also developed RIGS for “how to be with others” step by step. This had effects on the participants’ self-esteem and ability to communicate, as they were able to receive more effective guidance thanks to their having built up better-consolidated internal working models for interactions. Thus, their skills became better and more deeply integrated within the children, which made it easier for them to generalize their experiences.

We chose the research methods that we used because the form of treatment is new and emerged in a clinical setting where it takes time to test and research new ideas. The advantage was that the therapists could follow the children’s development even after treatment ended, since they are enrolled at the habilitation service center until the age of 18. What we found was that virtually all the children have managed with few habilitation interventions even for several years after treatment. Some children underwent follow-up cognitive assessments, which have shown that their development has continued to progress. In this study, we have only followed one child, which does not allow a generalizability regarding how children with ASD as a whole respond to this method, but through its longitudinal and multi-method approach we can reveal in more detail how a child with autism spectrum disorder responds to this treatment method. Despite the weaknesses, there are factors that increase the method’s reliability. The children’s developmental stages were discussed jointly among the therapists after each treatment session before writing them down as process notes. One therapist’s view of a child’s behavior could be contradicted or confirmed by another therapist, and a consensus was sought to get the most likely picture of events. We also analyzed the children’s developmental stages regularly together with the supervisor. Likewise, the children’s developmental steps were triangulated with the parents’ perspectives and that of the clinical and research supervisors. To return to William, reliability was strengthened by the results coming from three different sources—therapists, parents, and test results—all of which paint a consistent picture of the skills that he had developed.

One important question that the study cannot answer is how Williams’ development would have unfolded had he not received this treatment. We do know, however, that before the start of treatment he had received some stimulation via the preschool (with his own pedagogical resource staff member) and at school, and that his parents had also had time to receive some support via the habilitation center; for example, support for encouraging William’s cognitive development through picture books. Despite this, Williams’ performance on the tests before treatment was at a very low level, especially with regard to verbal ability, which showed a clear positive change after treatment. He had also reached the chronological age of eight at the time of the pre-treatment administration of the WPPSI-IV, which is a relatively advanced point to expect catch-up effects to be achieved. In one of the two verbal subtests—similarities—William performed significantly better after treatment than before. This subtest measures concept formation and more abstract thinking, areas that are close to mentalization. We can therefore reasonably believe that the treatment actually did affect Williams’ development in a positive way compared to if he had not received the treatment.

Are William’s results unique to him, or is it likely that more children with autism and intellectual disabilities could benefit from similar treatment protocols? Our purpose here was to investigate how autistic traits in a child can be managed within a treatment program such as COMSI^®^. This study, therefore, needs to be followed by a study that examines how COMSI^®^ works in a larger group of children with autism, including a control group, and if possible, randomization. A larger study with a control group would be able to yield answers to this question at a group level. However, it is not easy to match treatment and control groups: due to the multifaceted and specific difficulties many children with autism have, it is still not always easy to know which children will respond well to which treatment. However, that is a problem that applies to research on all treatments for children with autism. William initially exhibited an autism profile with extensive difficulties in verbal communication, in addition to difficulties with social interaction. In his early years, William had major behavioral problems and had had difficulty adapting to his first years of preschool. He tired very quickly and then shut himself off. For William, it was probably very valuable to receive a form of treatment rooted in physical activity and self-agency and based on his own interests and initiative, in which the approach was relational. He received a lot of help to build trust both in himself and others, and in this way his self-development improved. It is not uncommon for children and adolescents with autism to have negative experiences in their early socialization process and adaptation to group settings such as preschool or school. These experiences can create major obstacles for their subsequent psychological development and learning. It is therefore important to adopt a pronounced reparative perspective in treatment for children with autism.

Even though the approach proposed here is quite adjustable to each child’s needs of support, it cannot suit all children. For instance, some children have sensibilities that mean they must handle information in a more structured and predictable way. There might also be children with autism who are afraid of animals, not interested in them, or not at all ready to engage socially.

## 5. Conclusions

There is a lack of detailed and longitudinal case reports of treatments for children with autism. In this study, we have sought to cover that need through a multi-method longitudinal case study of a nature-based treatment for William. What we particularly want to highlight in this case study is the following: Our interpretation is that COMSI ^®^ respects the child as it provides special opportunities for the child to find their very own opportunities for self-development, language, and self-respect. We find that it depends on the special kinds of vitality forms [32] that nature and animals convey, which the children with autism more easily can interpret and process. This gives the child a wealth of vitality and also the therapist unique opportunities to flexibly and gently support the child where occasions arise. With more effective mental processes such as these, some of the cognitive obstacles within autism, such as executive function, central coherence and mentalizing, might be diminished.

Our hope is that through this case study we contribute new perspectives and important knowledge that may benefit more children with autism in the future, especially those children who cannot easily learn and develop in more traditional treatment modalities that currently exist.

For thousands of years, humans have been adapted to interpret the behavior of animals and events in nature [70,108,109,110]. This could perhaps in the future also be a part of a new avenue of treatment to instill psychological development and wellbeing for people with autism.

## Data Availability

The data is not publicly available because it is not in accordance with the consent of the parents of the children regarding the use of process notes, observations, pictures, interviews or test results.

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
