# Peer review of "COMSI®—A Form of Treatment That Offers an Opportunity to Play, Communicate and Become Socially Engaged through the Lens of Nature—A Single Case Study about an 8-Year-Old Boy with Autism and Intellectual Disability"

_ijerph, 2022, doi:10.3390/ijerph192416399_

Round 1

Reviewer 1 Report

1. The references and citations do not follow the required format for the journal.

2. The references must be updated or excluded.  Many are far too old to provide any current knowledge related to our current understanding of this topic.  Example: Line 52 states "today it is common in treatment  . . . " but references are from 1989, 1995, 2000 and 2010 - not close to up to date.

3. General comment, again, new approaches cannot be cited with old references.

4. The theoretical foundation of the model is not described, just referenced, and then goes on to discuss (line 175) how the therapist work rather than the foundation of the model.  Another theory is also bought up in the process notes section, this should move to the theory section.

5. The nature and animal intervention portion is ok, again references are way too old, but then you go on to describe MMSCR, this would be in methods, not here.  You state this can be used to develop new theories, take this out as you are not developing a new theory.

6. Procedure needs to include all procedures.  There is no mention of parent interviews, what assessments are used when, etc.  General organization is lacking here.  Stay true to the case study, the other children involved in the treatment were not consented or a part of the study so do not mention this here, it is not relevant.

7. Assessment - please report the reliability and validity of the instruments used.  update references

8.  The history of each research method is not beneficial.

9. Process notes and parent description should be moved to results.

10.  Start kindergarten or preschool - this is confusing.

11.  Language is offensive "lack of cooperation" - reword to unable to complete due to developmental age

12. Progress during treatment needs more specific time parameters, maybe months, rather than "as time went on" or "toward the end" Time with therapists became longer - what does this mean 5 to 10 minutes, or 5 to 60 minutes.  Report needs to be much more concise.

13. Report of post treatment testing - The first paragraph (False Beliefs) is concise, easy to read, with pre and post tests. Follow this same format for Cognition and Languages - make is clear, pre-post specifics, not generalizations, one paragraph.

14.  Overall, I suggest this be much more concise and clear in formatting and delivery.  There is not need for the discussion of methods (first paragraph), you have already addressed this several times.  The next 2 paragraphs would belong in discussion or conclusion, not in methods.

Reviewer 2 Report

(1) The manuscript needs English proofreading.
(2) What is the limitation of the proposed approach?

Reviewer 3 Report

Dear Authors,

Congratulations on this very difficult research. The fact that it was presented in the form of a case example was very helpful in understanding the method in detail. My suggestions about the article are available in the comment boxes of the attached file.

I wish for convenience.

Round 2

Reviewer 1 Report

Excellent revisions, much stronger and consistent overall.

Line 53 - reorder references

And references 26 and 60 should be translated to English.

Author Response

Thank you for your valuable opinion on our manuscript! We have now made the changes you required concerning the references.

Best regards, The authors